# Household food insecurity is associated with greater autonomic dysfunction testing score in Latinos with type 2 diabetes

Angela Bermúdez-Millán[1], Richard Feinn[2], Rachel Lampert[3], Rafael Pérez-Escamilla[4], Sofia Segura-Pérez[5], Julie Wagner[6]*

1 Community Medicine and Health Care, School of Medicine, UConn Health, Farmington, CT, United States of America, 2 School of Medicine, Medical Sciences, Quinnipiac University, Hamden, CT, United States of America, 3 Department of Internal Medicine, Yale School of Medicine, New Haven, CT, United States of America, 4 Department of Social and Behavioral Sciences, Yale School of Public Health, New Haven, CT, United States of America, 5 Chief Program Officer, Hispanic Health Council, Hartford, CT, United States of America, 6 Behavioral Sciences and Community Health, School of Dental Medicine, UConn Health, Farmington, CT, United States of America

* juwagner@uchc.edu

## Abstract

### Aim

We examined household food insecurity (HFI) and autonomic nervous system (ANS) function in a subset of low-income Latinos with type 2 diabetes with data from a stress management trial.

### Methods

#### Inclusion

Latino or Hispanic, Spanish speaking, age less than 18 years, ambulatory status, type 2 diabetes more than 6 months, A1c less than 7.0%.

#### Exclusion

Pain or dysfunction in hands (e.g., arthritis) precluding handgrip testing; medical or psychiatric instability. HFI was assessed with the 6-item U.S. household food security survey module; with responses to > = 1 question considered HFI. An ANS dysfunction index was calculated from xix autonomic function tests which were scored 0 = normal or 1 = abnormal based on normative cutoffs and then summed. Autonomic function tests were: 1) 24-hour heart rate variability as reflected in standard deviation of the normal-to-normal (SDNN) heart rate acquired with 3-channel, 7-lead ambulatory electrocardiogram (Holter) monitors; 2) difference between the highest diastolic blood pressure (DBP) during sustained handgrip and the average DBP at rest; 3) difference between baseline supine and the minimal BP after standing up; and, from 24-hour urine specimens 4) cortisol, 5) normetanephrine, and, 6) metanephrine.

**Data Availability Statement:** The data underlying the results presented in the study are available at https://osf.io/6hbn7/.

**Funding:** Supported by the National Institute of Minority Health and Health Disparities (R01MD005879) to RPE and JW; the American Diabetes Association (7-13-TS-31) to JW; and, the Federation of American Societies for Experimental Biology to ABM (FASEB 2011 Postdoctoral Professional Development and Enrichment Award). The funders had no role in study design, data collection and analysis, decision to publish, or preparation of the manuscript.

## Results

Thirty-five individuals participated, 23 (65.7%) of them were women, age mean = 61.6 (standard deviation = 11.2) years, HbA1c mean = 8.5% (standard deviation = 1.6) and 20 participants (57.1%) used insulin. Twenty-two participants (62.9%) reported HFI and 25 (71.4%) had one or more abnormal ANS measure. Independent t-tests showed that participants with HFI had a higher ANS dysfunction index (mean = 1.5, standard deviation = 0.9) than patients who were food secure (mean = 0.7, standard deviation = 0.8), p = 0.02. Controlling for financial strain did not change significance. Total ANS index was not related to glycemia, insulin use or other socioeconomic indicators. In this sample, HFI was associated with ANS dysfunction. Policies to improve food access and affordability may benefit health outcomes for Latinos with diabetes.

## Introduction

Food insecurity is not uncommon. In 2021, 10.2 percent (13.5 million) of U.S. households were food insecure at some time during the year [1]. Food insecurity (FI) is the limited or uncertain availability of nutritionally adequate and safe foods or the limited or uncertain ability to acquire acceptable foods in socially acceptable ways (that is, without resorting to emergency food supplies, scavenging, stealing, or other coping strategies) [1]. In the general population, FI is associated with all-cause mortality and cardiovascular mortality [2]. Among people with diabetes, FI is associated with suboptimal glycemic control [3], severe hypoglycemia [4], emergency department visits [5], hospitalizations and healthcare costs [6].

One possible link between FI and poor diabetes outcomes is autonomic nervous system (ANS) dysfunction. The ANS regulates body temperature, sweating, digestion, blood pressure and heart rate. The ANS is divided into parasympathetic and sympathetic branches, based on anatomy and function. Autonomic imbalance manifests as parasympathetic underactivity, and/or sympathetic hyperactivity, with individual variation in the relative contribution of each. Parasympathetic function can be non-invasively measured with heart rate variability (HRV) [7]. HRV analysis quantifies the beat-to-beat changes in heart rate caused by changes in vagal activity. Low HRV predicts mortality in diabetes [8]. Reduced HRV is a strong predictor of mortality, particularly among people with diabetes [9]. Sympathetic tone can be measured with non-invasive functional 'bedside' assessments. For example, sympathetic function can be measured with blood pressure response to a variety of stimuli including standing and handgrip [10]. Whereas each test has a normal cutoff value, ideally several are performed in concert, giving fuller information about the state of the autonomic system [11]. Autonomic imbalance is common [12] and can progress to overt diabetic autonomic neuropathies which can be disabling and life threatening [13].

Relationships among FI and diabetic autonomic function are particularly germane for Latinos with diabetes. In the U.S., analyses examining trends in food insecurity from 2001 to 2016 found that food insecurity rates for both non-Hispanic black and Hispanic households were at least twice that of non-Hispanic white households [1]. Similarly, rates of type 2 diabetes are higher among Hispanics (11.8%) than among non-Hispanic whites (7.4%) [14]. Latinos are 50% more likely to die from diabetes than are non-Hispanic whites [15].

Currently, little is known about the relationship between FI and autonomic function in Latinos with diabetes. Elucidating this relationship may point to modifiable risk factors and potentially even policy implications to improve outcomes among Latinos with diabetes. To

address this gap, we conducted secondary data analysis of a stress management trial for Latinos with poorly controlled type 2 diabetes (Community Health Workers Assisting Latinos Manage Stress and Diabetes, or CALMSD). We hypothesized that (a) food insecurity would be associated with lower autonomic function, and (b) glucose, glycemic control and insulin use would not fully account for this relationship.

## Materials and methods

### Setting and sample

The CALMSD study [16] was a collaboration among UConn Health, Yale University, the Hispanic Health Council, and Hartford Hospital. This study was approved by UConn Health Institutional Review Board (approval #11-065-6) and the trial was registered at clinicaltrials. gov (NCT01578096). Data were collected between August 21, 2013 and June 30, 2017.

Study participants were recruited from the 'Brownstone Clinic' clinic, an outpatient clinic at Hartford Hospital, serving low-income patients with diabetes, approximately 80% of which were Latinos at the time of the study. Participants in the randomized trial were residents of Hartford, CT, USA, aged 18 or older, Latino or Hispanic, Spanish-speaking, ambulatory, with type 2 diabetes for 6 months duration or longer and most recent past year A1c>7.0. Chart review excluded patients from CALMSD for: medical instability or intensive medical treatment; bipolar disorder or thought disorder; or suicide attempt or psychiatric hospitalization in the past two years. Face-to-face screening excluded patients with substance use problems or those who were enrolled in another diabetes research study. A subset of participants was also invited to participate in this substudy after their completion of CALMSD. An additional exclusion criterion was pain or dysfunction in hands (e.g., arthritis) precluding handgrip testing.

A total of 121 individuals had baseline assessments for the CALMSD trial, n = 107 were randomized and n = 96 completed post assessments. Of them, n = 11 did not wear a Holter monitor, HRV data from n = 15 required >20% .20% interpolation and so were excluded, and 1 individual did not provide food security data. Of the remaining 69, n = 34 declined the substudy or were excluded due to hand dysfunction (e.g., arthritis) that precluded the handgrip test. As previously reported [17, 18] this yielded 35 participants for analysis.

### Procedures

Clinic patients who met the study criteria were referred to Hispanic Health Council to be informed about the parent study and, after deciding to participate, provide written informed consent. A total of 121 Latinos with T2DM completed face-to-face baseline surveys with trained bilingual/bicultural interviewers with remote REDCap system [19]. One-hundred and seven participants were randomized to either diabetes education (DE) or diabetes education plus group stress management (DE+SM) and n = 96 completed post-assessments.

Early morning assessments were performed by CHW research personnel [20] to obtain fasting blood and urine samples. Fasting blood samples were collected early morning (i.e., before 9 am) in the participants' homes from an antecubital vein, using sterile technique, by the study's phlebotomist and trained data collector. Samples were promptly transported by trained study personnel, compliant with HIPAA and biohazard regulations, in an ice chest to the UConn Health Clinical Research Center laboratory for processing and analysis. Participants collected and kept refrigerated all their urine over a 24-hour period which study staff transported samples to the university laboratory for analysis.

Participants underwent 7-lead, 3 channel 24-hour Holter monitoring. 24-h Heart rate variability. Ambulatory electrodcardiograms (ECGs or Holters) were recorded on GE Medical (Milwaukee, WI) Marquette Series 8500 direct (amplitude-modulated) recorders and scanned

by an experienced technician. ECG pads were placed on the chest and connected to a portable monitor attached to the belt or worn on a strap around the neck. Participants were instructed the equipment and were given a toll-free number to call if an electrode fell off. On the next day, the phlebotomist removed the Holter monitor, collected equipment, and compensated participants $10 for surveys and $10 for each biological assessment.

## Measures

**Demographic and socio-economic characteristics.** Demographic information included age, gender, marital status (partnered vs. not), number of years living in the United States, language spoken (English only vs. Spanish only vs. English and Spanish). Socio-economic information included income (≤ $1000 or ≥ $1,001 per month), education (high school graduate/GED or less than high school graduate/GED), and Supplemental Nutrition Assistant Program (SNAP) benefits program participation (yes vs. no). We measured self-reported financial strain over the past 12 months on a scale from 1 = "we have enough and we can save" to 4 = "we don't have enough and we have great difficulties" [21].

**Clinical characteristics.** Fasting plasma glucose was analyzed at the REDACTED Hospital Laboratory using the LXI R system by Beckman Coulter. Glycemic control over the past approximately 3 months was indicated by HbA1c. It was measured in the REDACTED clinical laboratory using high pressure liquid chromatography (HPLC). In this laboratory, HbA1c shows the following coefficients of variation for normal and high values: Level 1 mean = 5.51%, CV = 3.3 based on n = 320, and Level 2 mean = 9.01%, CV = 3.2 based on n = 304. Use of exogenous insulin was per self-report yes vs. no.

**Independent variable.** *Food security status.* Food security status was assessed using the well-validated 6-item U.S. Household Food Security Measurement Module, short form [22, 23], in our sample, Cronbach's alpha = 0.89. Sample items are: "The food that (I/we) bought just didn't last, and (I/we) didn't have money to get more"; and "Were you ever hungry but didn't eat because there wasn't enough money for food?" Items referenced the previous 12 months. Response options were "yes"; "every month or almost every month", "some months but not every month", and "only one month". The sum of affirmative responses produces a scale score (0–6). Higher scores indicate greater food insecurity. Recent evidence indicates people in households with "marginal food security", usually classified as food secure in the U.S. government's prevalence estimates, may also face an increased likelihood of impaired health and nutrition [24, 25]. For this reason, we defined scores = 0 as food secure and scores ≥1 as food insecure, including marginal, low and very low food security levels.

**Dependent variable.** Our primary outcome was the total autonomic dysfunction score, which was created by converting each of the six autonomic measures into normal/abnormal categories using established published cutoffs and then summing across the six categories. Our autonomic dysfunction score was modeled after the Composite Autonomic Severity Score (CASS) [26] which was designed for use among people with diabetes. We modified the CASS by excluding the Valsalva maneuver (i.e., a forceful attempt of exhalation against a closed airway) because it is contra-indicated in patients with retinopathy, and we did not screen for retinopathy. We also excluded other CASS measures that were not feasible in the home, i.e., the sudomotor axon reflex test (an uncomfortable test of approximately 1-h that requires special equipment), the HRV during deep breathing test and the heart rate response to standing test (both require precise marking of the digital Holter recording). Instead, we collected more feasible home-based ANS measures, i.e., 24-hour HRV and catecholamines.

*Heart rate variability.* 24-hour HRV was assessed as in our previous studies [27]. Holter recordings were scanned by an experienced technician. As described in the literature [28],

each tape was manually processed and edited. The R-R interval data file was edited to remove ectopics (irregular heartbeats that are usually benign) and noise, and gaps filled in by interpolated linear splines. Those Holters with > 20% interpolated segments or less than 18 total recorded hours were excluded from further analysis. In the time domain, we assessed the standard deviation of the normal-to-normal R- R interval (SDNN). Normal value is defined as 149 +/-39 [29].

*Blood pressure response to standing.* The participant's blood pressure (BP) is measured with a digital sphygmomanometer while lying down, and again 4 minutes after standing up unaided. The postural falling BP is taken as the difference in supine and the minimal blood pressure after standing up [30]. Normal values are decline in systolic < = 20 mmHg and diastolic < = 10 mmHg [31].

*Blood pressure response to sustained handgrip.* The maximum voluntary contraction is first determined using a handgrip dynamometer. Handgrip is then maintained at 30% of that maximum for as long as possible, up to 5 minutes. Blood pressure was measured three times before and at 1-minute intervals during handgrip. The result was expressed as the difference between the highest diastolic during handgrip and the mean of the 3 diastolics before handgrip began [30]. A normal value is >15 mmHg [31].

*Catecholamines.* Urinary cortisol, metanephrine, and normetanephrine were analyzed from 24-h urine samples at the REDACTED Clinical Research Center core lab. Because epinephrine has a half-life of only 2 minutes, we measured metanephrine, one of 2 inactive principal metabolites of epinephrine found in urine only, as well as normetanephrine. We used a bi-neph kit from Alpco Diagnostics. The cortisol assay has a normal range of 50–190ug/24hrs, sensitivity of 2.0 ng/mL, intra-assay CV% = 7.4, and interassay CV% = 7.8. The metanephrine assay has a normal range of <350 ug/day, sensitivity of 13 ng/mL, intra-assay CV% = 8.9, and interassay CV% = 12. The normetanephrine assay has a normal range of <600 ug/day, sensitivity of 23 ng/mL, intra-assay CV% = 17.5, and interassay CV% = 20.4.

## Statistical analysis

The autonomic dysfunction score was compared between food insecurity status groups using the nonparametric Mann-Whitney test and r was reported as an effect size. In addition, to examine the potential role of glycemic status, the groups were compared on A1c and blood glucose also using the Mann-Whitney test. To isolate the effects of food insecurity from other socioeconomic indicators, the groups were compared on education, employment, income, and financial strain. Analyses were conducted in SPSS v28 and 2-sided statistical significance was set at an alpha level of 0.05.

## Results

Sample demographic characteristics are shown in Table 1. Most participants were female, over 60 years of age, had less than a high school degree, were not employed, had a weekly income below $1000 and were single, HbA1c M = 8.5% (SD = 1.6) and n = 20 (57.1%) used insulin. There was no significant difference between treatment groups (DE vs DE+SM) on autonomic dysfunction index.

Among the six items assessing food insecurity 13 participants (37.1%) endorsed none and 6 participants (17.1%) endorse all six items, median = 2 and mean = 2.3 (SD = 2.3). For the six ANS measures the range of measurements meeting the cut-off for dysfunction ranged from zero (13 participants, 37.1%) to three (3 participants, 8.6%). No participant had four or more dysfunctions and the mean of the sum score of dysfunctions was 1.2 (SD = 0.9).

**Table 1. Autonomic tests, normal values, and citations.**

| Measure | Value | Reference |
|---|---|---|
| Metanephrine | Normal <350 ug/day | Assay kit normal values, Alpco Diagnostics, Salem, NH |
| Normetanephrine | Normal <600 ug/day | Assay kit normal values, Alpco Diagnostics, Salem, NH |
| Cortisol | Normal 50–190 ug/day | Assay kit normal values, Alpco Diagnostics, Salem, NH |
| BP response to handgrip | Difference between the highest DBP during the handgrip and the average DBP at rest. Normal >15 mmHg. | Zygmunt 2010 |
| BP response to standing | BP 4 minutes after standing. Expressed as the difference between the baseline supine and the minimal BP after standing up. Normal = decline in SBP < = 20 mmHg and DBP < = 10 mmHg. | Zygmunt 2010 |
| 24-h HRV | SDNN normal = 149 +/-39 | HRV task force 1996 |

BP = blood pressure

D = diastolic

S = Systolic

HRV = Heart rate variability

SDNN = Standard deviation of the normal-to-normal heart beat

Table 2 compares food secure and food insecure participants on autonomic dysfunction, A1c, and glucose. There was a statistically significant difference on autonomic dysfunction (p = .023) and the effect was of clinical and public health significance (Fig 1). There was no significant difference between FI groups on A1c, blood glucose, or insulin use.

There was a significant association between FI and financial strain (p < .001). When financial strain was added as a covariate to the model comparing FI groups on autonomic score, the

**Table 2. Participant demographic characteristics.**

| | Frequency | Percentage |
|---|---|---|
| **Gender** | | |
| Female | 23 | 65.7% |
| Male | 12 | 34.3 |
| **Age** | | |
| Mean ± SD | 61.3 ± 11.2 | |
| **Marital Status** | | |
| Single | 23 | 65.7% |
| Partnered | 12 | 34.3 |
| **Education Level** | | |
| Less than High School | 26 | 74.3% |
| High School or More | 9 | 25.7 |
| **Employment Status** | | |
| Not Working | 30 | 85.7% |
| Working | 5 | 14.3 |
| **Monthly Income** | | |
| $1000 or less | 21 | 60.0% |
| $1001 - $2000 | 11 | 31.5 |
| Over $2000 | 3 | 8.6 |
| **Financial Strain** | | |
| Mean ± SD | 2.5 (0.8) | |

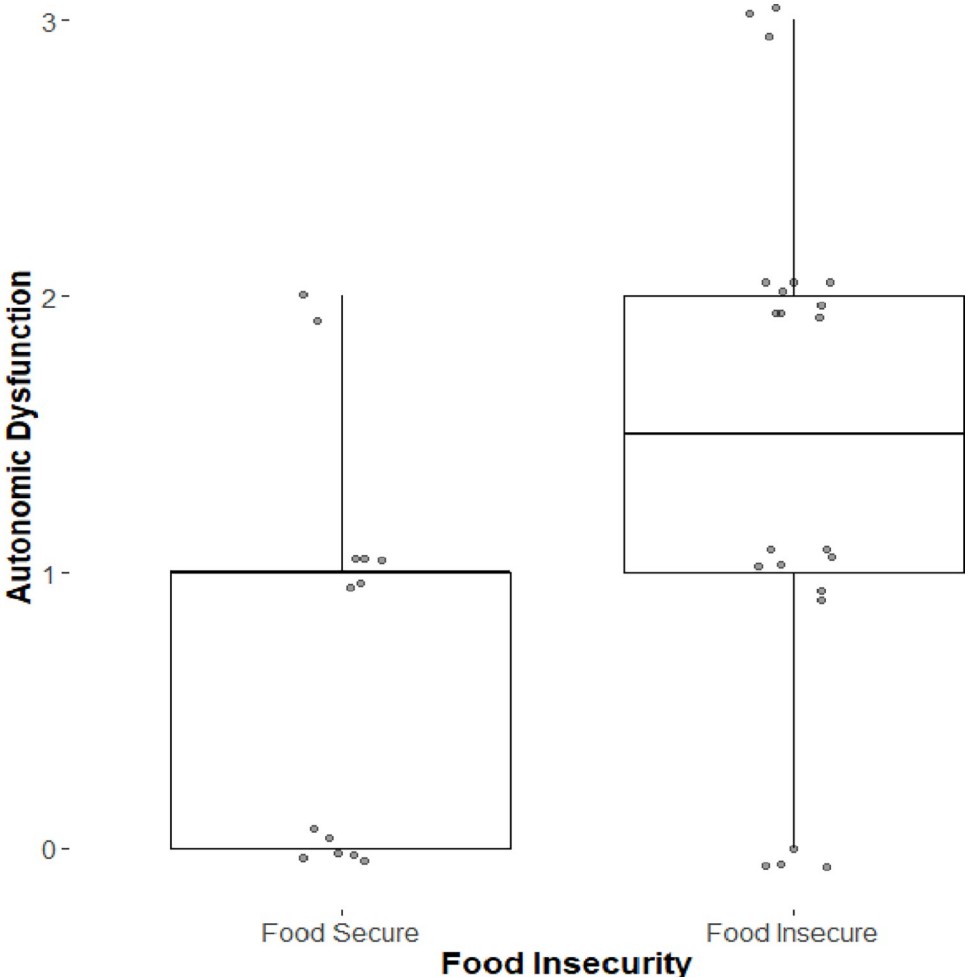

**Fig 1. Boxplot of number of autonomic dysfunctions by food insecurity status.** Food insecurity was assessed with the 6-item US Household Food Security Module [22].

difference between FI groups on autonomic score remained significant (p = .022). There was no significant difference between FI groups on education, income, or employment.

## Discussion

The main finding is that, among Latinos with type 2 diabetes, individuals experiencing household food insecurity demonstrated more autonomic dysfunction than their food secure counterparts. The effect size for the difference was small-to-moderate, not unlike standard risk factors such as smoking [32], hence of clinical and public health importance. That subjective financial strain did not account for FI suggests that FI is a distinctive socioeconomic indicator that poses a unique risk. This study is the first to examine links between FI and composite autonomic function. It aligns with and expands the literature regarding the deleterious association between FI and long-term health outcomes for people with diabetes [3, 33].

Our study findings have important implications for human health. Autonomic imbalance is associated with increased cardiovascular and all-cause mortality [9] and may be especially hazardous in people with diabetes (i.e., confers approximately doubled risk). Diabetic autonomic neuropathy may affect many organ systems throughout the body (e.g., gastrointestinal,

genitourinary, and cardiovascular). The latter, diabetic cardiac autonomic neuropathy, is the most serious [12] and is a risk for heart disease [34], cardiovascular events [35] including silent myocardial infarction, stroke [36] and mortality [37].

This exploratory study did not examine lifestyle-related mechanisms by which food insecurity and autonomic neuropathy may be associated. Quantity and quality of diet may directly impact health of the autonomic nervous system through macro and micronutrient deficiencies, or it may do so indirectly via overweight/obesity, insulin resistance and poor glycemic control in part related to the large consumption of ultra-processed foods and sugar sweetened beverages in the U.S. [38] and other high-income countries [39]. Worry about food and feeding the household is a chronic mental stressor [40], which may also negatively impact autonomic function [41]. Alternatively, individuals with autonomic neuropathy may experience more disability which places them at financial risk for food insecurity. Common factors, such as age and diabetes duration, may influence risk for both FI and autonomic dysfunction and we hypothesize that very likely they several factors synergize to strongly impact this key functional outcome.

## Limitations and future directions

There are several important limitations of this study. First, as mentioned, cross-sectional data cannot establish temporal precedence or causality between HFI and autonomic dysfunction. Prospective studies would allow for testing modifiable lifestyle causal mechanisms underlying relationships between HFI (and the extent to which respondents experience its most severe form, i.e., hunger) on autonomic function. Cross-sectional data were collected at post-treatment assessment from both control (DE) and treatment groups (DE+SM); however, these groups did not differ on autonomic score and comparisons between groups did not indicate that combining groups was inacceptable. Our sample was small, so null findings between FI and other variables of interest which differ from past studies (e.g., glycemic control [3], income [1]) need to be interpreted with caution due to potential type II error. Data were collected several years ago, though there is no particular reason to suspect that updated data collection would yield different results. The sample was low-income, Latino and findings may not generalize to other populations. Limitations are generally outweighed by the study's strengths including detailed functional challenge tests and a vulnerable and hard-to-reach clinical sample conducted in the participant's homes by carefully selected, thoroughly trained, and closely supervised CHWs [20].

Future studies should examine hypoglycemia unawareness in food insecure individuals. Autonomic dysfunction is the central factor in diabetic hypoglycemia unawareness. Autonomic dysfunction blunts the normal counter-regulatory hormone responses to low and dropping glucose levels. Patients with hypoglycemia unawareness cannot experience or detect the symptoms such as sweating, shaking, and weakness that indicate need for treatment with fast acting carbohydrate. Thus, they are at increased risk for severe low blood glucose (defined as the need for assistance to eat). People with hypoglycemia unawareness are also less likely to be awakened from sleep when hypoglycemia occurs at night. Impaired awareness of hypoglycemia is a common problem [42]. Hypoglycemia unawareness may be one reason, in addition to lack of food, for the elevated rates of severe hypoglycemia and emergency department visits observed in food insecure individuals [4]. Future mixed-methods studies should explore in depth hypoglycemia unawareness among food insecure individuals with diabetes.

## Conclusions

In this study, Latinos with type 2 diabetes who are food insecure showed lower autonomic function compared to their counterparts who are food secure, even after controlling for

financial strain. Those individuals who endorsed household food insecurity demonstrated more autonomic dysfunction (HRV, BP response to physical challenge, and catecholemines) than their food secure counterparts, with small to moderate effect sizes.

The U.S. national policy on hunger, nutrition and health aims to improve food access and affordability, empower consumers, support physical activity, integrate nutrition and public health and enhance research. These efforts, in concert with diabetes education and counseling that takes into account the social determinants of health [43], could yield downstream benefits for autonomic tone and diabetes outcomes.

## Author Contributions

**Conceptualization:** Julie Wagner.

**Data curation:** Sofia Segura-Pérez.

**Formal analysis:** Richard Feinn.

**Funding acquisition:** Angela Bermúdez-Millán, Rafael Pérez-Escamilla, Julie Wagner.

**Investigation:** Angela Bermúdez-Millán, Rafael Pérez-Escamilla, Julie Wagner.

**Methodology:** Rachel Lampert.

**Project administration:** Rafael Pérez-Escamilla, Sofia Segura-Pérez, Julie Wagner.

**Software:** Rachel Lampert.

**Supervision:** Angela Bermúdez-Millán, Rachel Lampert, Rafael Pérez-Escamilla, Sofia Segura-Pérez.

**Writing – original draft:** Julie Wagner.

**Writing – review & editing:** Angela Bermúdez-Millán, Richard Feinn, Rachel Lampert, Rafael Pérez-Escamilla, Sofia Segura-Pérez.

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
