## [Decision Letter · Decision Letter 0]

31 Oct 2023

PONE-D-23-25306Household food insecurity is associated with greater autonomic dysfunction testing score in Latinos with type 2 diabetesPLOS ONE

Dear Dr. Wagner,

Thank you for submitting your manuscript to PLOS ONE. After careful consideration, we feel that it has merit but does not fully meet PLOS ONE’s publication criteria as it currently stands. Therefore, we invite you to submit a revised version of the manuscript that addresses the points raised during the review process.

EDITOR'S COMMENTSI am pleased to inform you that two anonymous reviewers have reviewed your manuscript and you are given the opportunity to attend to the comments/suggestions as early as you can. Please submit your revised manuscript by Dec 15 2023 11:59PM. If you will need more time than this to complete your revisions, please reply to this message or contact the journal office at plosone@plos.org. Please include the following items when submitting your revised manuscript:A rebuttal letter that responds to each point raised by the academic editor and reviewer(s). You should upload this letter as a separate file labeled 'Response to Reviewers'.A marked-up copy of your manuscript that highlights changes made to the original version. You should upload this as a separate file labeled 'Revised Manuscript with Track Changes'.An unmarked version of your revised paper without tracked changes. You should upload this as a separate file labeled 'Manuscript'.

We look forward to receiving your revised manuscript.

Kind regards,

Olutosin Ademola Otekunrin, PhD

Academic Editor

PLOS ONE

- https://doi.org/10.2337/db19-2284-PUB

- https://doi.org/10.1093/cdn/nzaa043_013

In your revision ensure you cite all your sources (including your own works), and quote or rephrase any duplicated text outside the methods section. Further consideration is dependent on these concerns being addressed.

4. Please ensure that you include a title page within your main document. You should list all authors and all affiliations as per our author instructions and clearly indicate the corresponding author.

5. Please amend your authorship list in your manuscript file to include authors Julie Wagner, Angela Bermudez-Millan, Richard Feinn, Rachel Lampert, Rafael Perez-Escamilla and Sofia Segura-Perez.

6. Please include a separate caption for each figure in your manuscript.

7. Please include your tables as part of your main manuscript and remove the individual files. Please note that supplementary tables (should remain/ be uploaded) as separate "supporting information" files.

Reviewers' comments:

Reviewer's Responses to Questions

**Comments to the Author**

1. Is the manuscript technically sound, and do the data support the conclusions?

Reviewer #1: Yes

Reviewer #2: Partly

2. Has the statistical analysis been performed appropriately and rigorously? 

Reviewer #1: Yes

Reviewer #2: Yes

3. Have the authors made all data underlying the findings in their manuscript fully available?

Reviewer #1: Yes

Reviewer #2: No

4. Is the manuscript presented in an intelligible fashion and written in standard English?

Reviewer #1: Yes

Reviewer #2: Yes

5. Review Comments to the Author

Reviewer #1: The study examines household food insecurity (HFI) and autonomic nervous system (ANS) function in a subset of low-income Latinos with type 2 diabetes. The topic is relevant however, the authors have to address these comments before granting any publication consideration.

1) The authors should incorporate the summary of their policy implication(s) into the abstract.

2) The authors should make clear the contribution(s) of the study.

3) The authors should compare their findings presented in the results section with previous studies and where possible give reasons to support their outcome (example, explain why (1) there was no significant difference between FI groups on A1c, blood glucose, or insulin use; (2) there was no significant difference between FI groups on education, income, or employment)

4) The authors should incorporate their findings into the conclusion section

Reviewer #2: Reviewer’s comments

The study investigated the nexus between household food insecurity (HFI) and autonomic nervous system (ANS) function in a subset of low-income Latinos with type 2 diabetes with data from a stress management trial. Generally, the article is well-written. However, the authors need to improve it. Below are my suggestions and comments.

Abstract

Methods: Present the methods adopted in sentences; avoid the use of symbol where necessary for easy understanding by the readers.

Results: Present the results in sentences; avoid the use of symbol where necessary for easy understanding by the multidisciplinary readers.

Introduction

The introduction is well-written. However, the authors can improve this section by clearly showing the research gap and how their study contributes to literature. In addition, they should include how the study is important to justify the current study.

Paragraph 2: define FI in the first use.

Results

The sentence “A total of 121 individuals had baseline assessments for the PARENT STUDY trial, 182 n=107 were randomized and n=96 completed post assessments. Of them, n = 11 did not wear 183 a Holter monitor, HRV data from n = 15 required >20% .20% interpolation and so were 184 excluded, and 1 individual did not provide food security data. Of the remaining 69, n=34 185 declined the substudy or were excluded due to hand dysfunction (e.g., arthritis) that precluded 186 the handgrip test. This yielded 35 participants for analysis” would be better under methods.

Discussion

The authors need to further discuss their findings by clearly given implications of each finding and link them with earlier studies that are in line or against their findings.

Conclusion

The authors need to have few sentences to conclude the paper in addition to what they have in this section. What the authors currently have under this section is recommendation, which should come after drawing inferences from the study.

6. PLOS authors have the option to publish the peer review history of their article (what does this mean?). If published, this will include your full peer review and any attached files.

Reviewer #1: No

Reviewer #2: No

---

## [Author Response · Author response to Decision Letter 0]

27 Nov 2023

To: Olutosin Ademola Otekunrin, PhD, Academic Editor, PLOS ONE

From: Julie Wagner

Date: 11/4/23

Re: PONE-D-23-25306 “Household food insecurity is associated with greater autonomic dysfunction testing score in Latinos with type 2 diabetes” 

We thank the editor and reviewers for their careful reviews. We were pleased with their overall positive impression of the manuscript. Below we respond to comments in bold font for easy spotting. We have made all the changes suggested by the reviewers. With their help, we have fashioned an even stronger manuscript. 

Comments from the Editor:

- https://doi.org/10.2337/db19-2284-PUB

- https://doi.org/10.1093/cdn/nzaa043_013

In your revision ensure you cite all your sources (including your own works), and quote or rephrase any duplicated text outside the methods section. Further consideration is dependent on these concerns being addressed.

We now cite these 2 published abstracts from conference presentations. We have also rephrased duplicated text that occurs outside the methods section. Duplicate text within the methods section has been mostly unrevised. 

Immediately below is a link to the data repository. Thank you for updating our data availability statement on your behalf.

https://osf.io/6hbn7/

4. Please ensure that you include a title page within your main document. You should list all authors and all affiliations as per our author instructions and clearly indicate the corresponding author.

The title page has been added to the main document.

5. Please amend your authorship list in your manuscript file to include authors Julie Wagner, Angela Bermudez-Millan, Richard Feinn, Rachel Lampert, Rafael Perez-Escamilla and Sofia Segura-Perez.

These individuals are now included in the authorship list on the title page.

6. Please include a separate caption for each figure in your manuscript.

We now include a figure caption.

7. Please include your tables as part of your main manuscript and remove the individual files. Please note that supplementary tables (should remain/ be uploaded) as separate "supporting information" files.

Tables are now included in the main manuscript. We have no supplementary files. 

Reviewers' comments:

Comments to the Author

1. Is the manuscript technically sound, and do the data support the conclusions?

Reviewer #1: Yes

Reviewer #2: Partly

2. Has the statistical analysis been performed appropriately and rigorously? 

Reviewer #1: Yes

Reviewer #2: Yes

3. Have the authors made all data underlying the findings in their manuscript fully available?

Reviewer #1: Yes

Reviewer #2: No

Here is a link to the data repository:

https://osf.io/6hbn7/

4. Is the manuscript presented in an intelligible fashion and written in standard English?

Reviewer #1: Yes

Reviewer #2: Yes

Reviewer #1 Comments:

The study examines household food insecurity (HFI) and autonomic nervous system (ANS) function in a subset of low-income Latinos with type 2 diabetes. The topic is relevant however, the authors have to address these comments before granting any publication consideration.

1) The authors should incorporate the summary of their policy implication(s) into the abstract.

The last sentence of the abstract incorporates policy implications.

2) The authors should make clear the contribution(s) of the study.

The discussion section now makes clear the contributions of the study.

3) The authors should compare their findings presented in the results section with previous studies and where possible give reasons to support their outcome (example, explain why (1) there was no significant difference between FI groups on A1c, blood glucose, or insulin use; (2) there was no significant difference between FI groups on education, income, or employment).

This is now presented in the Limitations section.

4) The authors should incorporate their findings into the conclusion section.

The findings are incorporated into the conclusion.

Reviewer #2: Reviewer’s comments

The study investigated the nexus between household food insecurity (HFI) and autonomic nervous system (ANS) function in a subset of low-income Latinos with type 2 diabetes with data from a stress management trial. Generally, the article is well-written. However, the authors need to improve it. Below are my suggestions and comments.

Abstract

Methods: Present the methods adopted in sentences; avoid the use of symbol where necessary for easy understanding by the readers.

Results: Present the results in sentences; avoid the use of symbol where necessary for easy understanding by the multidisciplinary readers.

We now report methods and results in sentences without the use of symbols or statistical abbreviations.

Introduction

The introduction is well-written. However, the authors can improve this section by clearly showing the research gap and how their study contributes to literature. In addition, they should include how the study is important to justify the current study.

We now state the knowledge gap and how our study addresses it. 

Paragraph 2: define FI in the first use. 

We define FI in the first use (in the first paragraph).

Results

The sentence “A total of 121 individuals had baseline assessments for the PARENT STUDY trial, 182 n=107 were randomized and n=96 completed post assessments. Of them, n = 11 did not wear 183 a Holter monitor, HRV data from n = 15 required >20% .20% interpolation and so were 184 excluded, and 1 individual did not provide food security data. Of the remaining 69, n=34 185 declined the substudy or were excluded due to hand dysfunction (e.g., arthritis) that precluded 186 the handgrip test. This yielded 35 participants for analysis” would be better under methods.

We have moved this to the methods section.

Discussion

The authors need to further discuss their findings by clearly given implications of each finding and link them with earlier studies that are in line or against their findings.

We have added this to the end of the first paragraph of the discussion.

Conclusion

The authors need to have few sentences to conclude the paper in addition to what they have in this section. What the authors currently have under this section is recommendation, which should come after drawing inferences from the study.

We have added a sentence with a clear, concise conclusion.

---

## [Decision Letter · Decision Letter 1]

12 Dec 2023

PONE-D-23-25306R1Household food insecurity is associated with greater autonomic dysfunction testing score in Latinos with type 2 diabetesPLOS ONE

Dear Dr. Wagner,

Thank you for submitting your manuscript to PLOS ONE. After careful consideration, we feel that it has merit but does not fully meet PLOS ONE’s publication criteria as it currently stands. Therefore, we invite you to submit a />==============================

ACADEMIC EDITOR: I am pleased to inform you that your manuscript has been reviewed by two anonymous reviewers. Kindly attend to the comments/suggestions of reviewer 1 as early as you can.==============================

We look forward to receiving your revised manuscript.

Kind regards,

Olutosin Ademola Otekunrin

Academic Editor

PLOS ONE

Journal Requirements:

Reviewers' comments:

Reviewer's Responses to Questions

Comments to the Author

1. If the authors have adequately addressed your comments raised in a previous round of review and you feel that this manuscript is now acceptable for publication, you may indicate that here to bypass the “Comments to the Author” section, enter your conflict of interest statement in the “Confidential to Editor” section, and submit your "Accept" recommendation.

Reviewer #1: (No Response)

Reviewer #2: All comments have been addressed

2. Is the manuscript technically sound, and do the data support the conclusions?

Reviewer #1: Yes

Reviewer #2: Yes

3. Has the statistical analysis been performed appropriately and rigorously? 

Reviewer #1: Yes

Reviewer #2: Yes

4. Have the authors made all data underlying the findings in their manuscript fully available?

Reviewer #1: Yes

Reviewer #2: No

5. Is the manuscript presented in an intelligible fashion and written in standard English?

Reviewer #1: Yes

Reviewer #2: Yes

6. Review Comments to the Author

Reviewer #1: The authors have tried to address all the comments. However, the authors need to improve the conclusion section by incorporating the summary of their findings.

Reviewer #2: I do not have further comments. My earlier comments have been addressed. The article is generally good.

7. PLOS authors have the option to publish the peer review history of their article (what does this mean?). If published, this will include your full peer review and any attached files.

Do you want your identity to be public for this peer review? For information about this choice, including consent withdrawal, please see our Privacy Policy.

Reviewer #1: No

Reviewer #2: Yes: Ridwan Mukaila

---

## [Author Response · Author response to Decision Letter 1]

4 Jan 2024

To: Olutosin Ademola Otekunrin, PhD, Academic Editor, PLOS ONE

From: Julie Wagner

Date: 1/4/24

Re: R2 PONE-D-23-25306 “Household food insecurity is associated with greater autonomic dysfunction testing score in Latinos with type 2 diabetes” 

Thank you for the second round of reviews. The reviews are below with our responses in bold font for easy spotting. The reviews request only one minor edit to the conclusion, which we have made.

1. If the authors have adequately addressed your comments raised in a previous round of review and you feel that this manuscript is now acceptable for publication, you may indicate that here to bypass the “Comments to the Author” section, enter your conflict of interest statement in the “Confidential to Editor” section, and submit your "Accept" recommendation.

Reviewer #1: (No Response)

Reviewer #2: All comments have been addressed

2. Is the manuscript technically sound, and do the data support the conclusions?

Reviewer #1: Yes

Reviewer #2: Yes

3. Has the statistical analysis been performed appropriately and rigorously? 

Reviewer #1: Yes

Reviewer #2: Yes

4. Have the authors made all data underlying the findings in their manuscript fully available?

Reviewer #1: Yes

 Reviewer #2: No 

Reviewer #2 is mistaken. We provided a link to a data repository, as noted by Reviewer #1. It is on the title page.

5. Is the manuscript presented in an intelligible fashion and written in standard English?

Reviewer #1: Yes

Reviewer #2: Yes

6. Review Comments to the Author

Reviewer #1: The authors have tried to address all the comments. However, the authors need to improve the conclusion section by incorporating the summary of their findings.

We now further incorporate the summary of our findings in the conclusion.

Reviewer #2: I do not have further comments. My earlier comments have been addressed. The article is generally good

---

## [Decision Letter · Decision Letter 2]

11 Jan 2024

Household food insecurity is associated with greater autonomic dysfunction testing score in Latinos with type 2 diabetes

PONE-D-23-25306R2

Dear Dr. Wagner,

We’re pleased to inform you that your manuscript has been judged scientifically suitable for publication and will be formally accepted for publication once it meets all outstanding technical requirements.

Kind regards,

Olutosin Ademola Otekunrin

Academic Editor

PLOS ONE

Additional Editor Comments (optional):

Reviewers' comments:

Reviewer's Responses to Questions

**Comments to the Author**

1. If the authors have adequately addressed your comments raised in a previous round of review and you feel that this manuscript is now acceptable for publication, you may indicate that here to bypass the “Comments to the Author” section, enter your conflict of interest statement in the “Confidential to Editor” section, and submit your "Accept" recommendation.

Reviewer #1: All comments have been addressed

Reviewer #2: All comments have been addressed

2. Is the manuscript technically sound, and do the data support the conclusions?

Reviewer #1: Yes

Reviewer #2: Yes

3. Has the statistical analysis been performed appropriately and rigorously? 

Reviewer #1: Yes

Reviewer #2: Yes

4. Have the authors made all data underlying the findings in their manuscript fully available?

Reviewer #1: Yes

Reviewer #2: Yes

5. Is the manuscript presented in an intelligible fashion and written in standard English?

Reviewer #1: Yes

Reviewer #2: Yes

6. Review Comments to the Author

Reviewer #1: The authors have addressed all my comments. The manuscript is now in a good state for publication.

Reviewer #2: I do not have further comments. My earlier comments have been addressed. The article is now generally okay.

7. PLOS authors have the option to publish the peer review history of their article (what does this mean?). If published, this will include your full peer review and any attached files.

Reviewer #1: No

Reviewer #2: No

---

## [Editor Report · Acceptance letter]

14 Feb 2024

PONE-D-23-25306R2 

PLOS ONE

Dear Dr. Wagner, 

I'm pleased to inform you that your manuscript has been deemed suitable for publication in PLOS ONE. Congratulations! Your manuscript is now being handed over to our production team.

Kind regards, 

on behalf of

Dr. Olutosin Ademola Otekunrin 

Academic Editor

PLOS ONE